# The Role of the Bacterial Muramyl Dipeptide in the Regulation of GLP-1 and Glycemia

**DOI:** 10.3390/ijms21155252

**Published:** 2020-07-24

**Authors:** Laura Williams, Amal Alshehri, Bianca Robichaud, Alison Cudmore, Jeffrey Gagnon

**Affiliations:** Department of Biology, Laurentian University, 935 Ramsey Lake Road, Sudbury, ON P3E 2C6, Canada; lwilliams1@laurentian.ca (L.W.); aalshehri2@laurentian.ca (A.A.); brobichaud@laurentian.ca (B.R.); Aocudmore@eastlink.ca (A.C.)

**Keywords:** muramyl dipeptide, postbiotic, gut hormone, NOD2, glucagon-like peptide-1

## Abstract

The host’s intestinal microbiota contributes to endocrine and metabolic responses, but a dysbiosis in this environment can lead to obesity and insulin resistance. Recent work has demonstrated a role for microbial metabolites in the regulation of gut hormones, including the metabolic hormone, glucagon-like peptide-1 (GLP-1). Muramyl dipeptide (MDP) is a bacterial cell wall component which has been shown to improve insulin sensitivity and glucose tolerance in diet-induced obese mice by acting through the nucleotide oligomerization domain 2 (NOD2) receptor. The purpose of this study was to understand the effects of MDP on GLP-1 secretion and glucose regulation. We hypothesized that MDP enhances glucose tolerance by inducing intestinal GLP-1 secretion through NOD2 activation. First, we observed a significant increase in GLP-1 secretion when murine and human L-cells were treated with a fatty acid MDP derivative (L18-MDP). Importantly, we demonstrated the expression of the NOD2 receptor in mouse intestine and in L-cells. In mice, two intraperitoneal injections of MDP (5 mg/kg body weight) caused a significant increase in fasting total GLP-1 in chow-fed mice, however this did not lead to an improvement in oral glucose tolerance. When mice were exposed to a high-fat diet, they eventually lost this MDP-induced GLP-1 release. Finally, we demonstrated in L-cells that hyperglycemic conditions reduce the mRNA expression of NOD2 and GLP-1. Together these findings suggest MDP may play a role in enhancing GLP-1 during normal glycemic conditions but loses its ability to do so in hyperglycemia.

## 1. Introduction

Glucagon-like peptide-1 (GLP-1) is a gastrointestinal hormone secreted from the enteroendocrine L-cell upon food consumption [1,2]. Although GLP-1 has numerous metabolic and protective properties, this hormone is best known for its action on pancreatic β-cells as it enhances glucose-dependent insulin secretion [3,4]. As a result, GLP-1-based therapeutics including GLP-1R agonists, have become an important part of improving glycemia for individuals with chronic metabolic diseases such as obesity and Type 2 Diabetes (T2D) [5].

In humans, L-cells are primarily found in the intestinal crypts of the ileum and colon [6]. Due to the proximity of these cells in relation to the intestinal microbiota, research has sought to define the impacts of the intestinal bacterial community on the development of metabolic diseases through GLP-1 [7,8]. Changes in diversity and richness of the microbial community are associated with the development of obesity and T2D [9,10]. As a consequence, bacterial metabolite production is likely altered and may contribute to host metabolism [11,12]. Therefore, manipulating the intestinal microbial makeup (probiotics and prebiotics) or administering microbial metabolites (postbiotics) in order to regulate host metabolism via gut hormone release is an attractive therapeutic approach [12,13].

Bacterial metabolites including short chain fatty acids (SCFAs), indole, hydrogen sulfide, and secondary bile acids have all been shown to stimulate GLP-1 secretion in vitro and/or in vivo [14,15,16,17]. Furthermore, a 3-month intervention in humans with pasteurized *Akkermansia muciniphila* improved several metabolic parameters including insulin sensitivity and a reduction in the GLP-1 degrading enzyme, dipeptidyl peptidase 4 (DPP4) [18]. This suggests that gut microbial components are contributing to improved metabolic health.

Although elevated concentrations of bacterial lipopolysaccharide are closely associated with obesity and insulin resistance, peptidoglycan motifs have shown the opposite effects [19]. All bacteria contain a cell wall comprised of peptidoglycan; alternating repeats and cross-linkages of *N*-acetylglucosamine (NAG) and *N*-acetylmuramic acid (NAM) bound to a short chain of amino acids which provides the structural strength for the bacterium [20]. Peptidoglycan motifs can be sensed in the host by nucleotide-binding oligomerization domain-like receptors (NLRs) [21]. Nucleotide-binding oligomerization domain-containing protein 2 (NOD2) is the cytoplasmic receptor for muramyl dipeptide (MDP). In immune cells, NOD2 activation causes pro-inflammatory cytokine release through the mitogen-activated protein kinase (MAPK) and NF-κB activation, thus contributing to host defense [22,23]. Furthermore, NOD2 contributes to intestinal microbial community composition as well as maintaining intestinal barrier function [24]. Although MDP stimulated NOD2 activation has been primarily examined in the innate immune system, Cavallari et al. (2017) [25] have demonstrated that MDP acts through NOD2 to improve insulin resistance and metabolic tissue inflammation in diet-induced obese mice. While it has been suggested that the transcription factor, interferon regulatory factor 4 (IRF4), is involved in the MDP-NOD2-induced insulin sensitizing effects during obesity, the precise mechanism of MDP’s beneficial effects has yet to be elucidated. We hypothesize that MDP improves glycemia through the regulation of GLP-1 secretion and action.

This study aims to understand the direct effects of MDP on GLP-1 secretion and glucose tolerance and how these effects are altered under hyperglycemic conditions.

## 2. Results

### 2.1. L18-MDP Stimulates GLP-1 In Vitro

In order to determine the effects of MDP on GLP-1 secretion, we incubated GLUTag cells with MDP (dosing ranging from 0.001–100 μg/mL) or control for a two-hour period (Figure 1A). While a slight increase at the 0.1 μg/mL dose was observed, there was no significance in the effect of MDP on GLP-1 secretion. To assist with cell uptake of MDP, we tested the fatty acid modified L18-MDP. GLUTag cells treated with both 5 and 10 μg/mL concentrations of L18-MDP had a significant increases in GLP-1 secretion (*p* < 0.0001 in post hoc analysis) with the higher concentration leading to a 3.29 ± 0.29-fold increase (from a basal secretion of 125 pg/mL) (Figure 1B). Conversely, the fatty acid modified nuclear oligomerization domain 1 (NOD1) agonist γ-D-Glu-mDAP (iE-DAP) had no effect on GLP-1 (Figure 1C). To make certain that the effects L18-MDP were not limited to murine L-cells, we treated the human NCI-H716 cells with this compound. In NCI-H716 cells, L18-MDP caused a similar GLP-1 fold-increase (3.85 ± 0.06 compared to control) at the 10 μg/mL concentration (*p* < 0.0001 in post hoc analysis, Figure 1D). Cell viability/toxicity of murine and human cells were examined using the neutral red absorbance assay and were not affected by L18-MDP (Figure 1E,F).

### 2.2. NOD2 is Expressed in L-cells

As MDP is the ligand for the NOD2 cytoplasmic receptor, we examined NOD2 expression in mouse ileum and in the L-cell lines. NOD2 and GLP-1 protein expression and distribution were detected in intestinal L-cells by performing double fluorescent immunohistochemistry in paraffin embedded mouse ileum sections (Figure 2A). The number of cells expressing NOD2, GLP-1, or both were counted using four different tissue samples where 3.6 ± 1.14 per 100 cells were positive for NOD2, 2 ± 0.37 per 100 cells were positive for GLP-1 and 1.6 ± 0.31 per 100 cells were positive for both NOD2 and GLP-1 proteins (Figure 2B). Moreover, NOD2 mRNA expression was detected in GLUTag cells (Figure 2C). Finally, we demonstrated the protein expression of NOD2 in NCI-H716 cells by western blot (Figure 2D). Western blotting for NOD2 in GLUTag cells did not yield a band of predicted size.

### 2.3. L18-MDP Causes a Slight Increase in p38 MAPK Phosphorylation

As previous studies have implicated the p38 MAPK signaling pathway (separately) in GLP-1 secretion and in NOD2 signaling, we examined p38 MAPK phosphorylation in MDP treated GLUTag cells. A modest but statistically significant increase in p38 MAPK phosphorylation was observed after the 10 min treatment. No increase in p38 MAPK phosphorylation was observed after the 20 min L18-MDP treatment (Figure 3).

### 2.4. The In Vivo Effects of MDP on GLP-1 Secretion in Chow-Fed Mice

To determine the effects of exogenous MDP on GLP-1 secretion in vivo, fasted chow-fed male and female mice were injected with MDP (5 mg/kg) or saline 30-min prior to blood sampling. Male and female results were combined as no statistically significant differences in the parameters measured were observed between sexes. After a single intraperitoneal (IP) injection of MDP, there was no significant difference in GLP-1 levels between MDP or saline-treated animals (Figure 4A). We then administered MDP (5 mg/kg) or saline, at the time of fasting (−16 h) as well as at the end of the overnight fast (−30 min). Here, MDP treatment lead to a 1.57 ± 0.13-fold increase in GLP-1 compared to control (from a basal secretion of 52.6 ± 6.44 pM) (*p* < 0.001, unpaired *t*-test) (Figure 4B). However, there were no improvements in glucose tolerance (Figure 4C) or insulin secretion (Figure 4D). Since we were interested in the role of MDP and GLP-1 mediating restored glucose tolerance, we repeated the experiment in high-fat diet-fed mice.

### 2.5. The In Vivo Effects of MDP on Glucose Tolerance in Obesogenic-Fed Mice

Mice were fed a 45% (calories from fat) high-fat diet (HFD) to drive hyperglycemic phenotypes in the animals. An oral glucose tolerance test (OGTT) was performed after two IP injections of MDP (5 mg/kg) or saline. No significant effect of MDP was detected on glucose tolerance (Figure 5A–C). Additionally, no effect was observed for GLP-1 and insulin secretion responses. While mice had elevated fasting glucose levels relative to baseline as the HFD continued (8.79 ± 0.50 mmol/L at 100 days, Figure 5D), the levels did not reach the level of marked hyperglycemia previously reported [25]. To induce pronounced levels of hyperglycemia, we treated a new cohort of mice with the high sugar and high fat Western diet (WD). After 70 days mice did exhibit a greater degree of fasting blood glucose levels (9.35 ± 0.42 mmol/L). Nevertheless, we did not observe any improvements in glucose tolerance in these animals when treated with the similar 2-injection MDP protocol (Figure 5E). Next, we altered the MDP injection schedule to provide three, once-daily IP injection of MDP (100 μg/mouse) or saline to hyperglycemic (10.47 ± 0.85 mmol/L) WD-fed mice. Again, no significant differences in glucose tolerance from MDP injections were observed in these animals (Figure 5F). Finally, we tested the NOD2 agonist, mifamurtide (50 µg/mouse), under the 2-injection protocol and found no improvements in glucose tolerance.

### 2.6. The Effects of MDP on Fasting GLP-1 Levels

As no effects of MDP on glucose tolerance were observed in both obesogenic models, we compared the effect of MDP injection on fasting GLP-1 and insulin levels in mice at different time points on the HFD. The initial significant increase in GLP-1 (*p* < 0.001 in post hoc analysis) in MDP treated mice was no longer observed even as soon as after 30 days on the obesogenic diet (Figure 6A). MDP had no effects on fasting insulin levels (Figure 6B).

### 2.7. The Effect of A Hyperglycemic Environment on NOD2 and GCG mRNA Expression

Since the in vivo experiments demonstrated a loss of GLP-1 response to MDP during the HFD (which occurs alongside elevated blood glucose), we examined the effects of a high glucose environment on *NOD2* and proglucagon (*GCG*, GLP-1 gene) mRNA expression in GLUTag cells. Cells were incubated in low glucose media (5 mmol/L) or high glucose media (25 mmol/L) for 24 or 48 h. We determined that *NOD2* expression was significantly reduced in the high glucose environment after the 48 h incubation (*p* < 0.05, in post-hoc analysis) (Figure 7A). In addition, the overall *GCG* expression was significantly reduced due to the high glucose treatment (*p* < 0.05, two-way ANOVA) (Figure 7B).

## 3. Discussion

In this study we examined the effects of MDP and derivatives on GLP-1 secretion in L-cells and animals. Initially we found that the acyl-modified MDP stimulates GLP-1 secretion. Next, we demonstrated the presence of the MDP receptor, NOD2, in the cell models as well as in mouse intestinal tissue. Since MDP was found to enhance GLP-1 secretion in vitro we next explored its effect in mice. While we did observe an MDP-induced enhancement in basal GLP-1 secretion, this did not lead to an improvement in glycemia. Furthermore, during an obesogenic diet, the MDP-induced enhancement in basal GLP-1 was lost. We demonstrated that this loss in effect may be due, in part, to the downward expression of *NOD2* under hyperglycemic conditions.

In order to stimulate GLP-1 secretion in vitro, the synthetic saturated stearoyl fatty acid derivative of MDP (L18-MDP) was required. Although L18-MDP is not naturally found in the gut, several protocols rely on the use of a fatty acid to facilitate the uptake of MDP [26,27]. We also demonstrated that a fatty acid modified NOD1 agonist (C12-iE-DAP) had no effects on GLP-1 secretion. This is important as long chain fatty acids are known ligands for the free fatty acid receptors on GLUTag cells. As no GLP-1 stimulation was observed with the NOD1 agonist, we can confirm that lipid modification, as well as NOD1 activation does not alter GLP-1 release. Our findings are in agreement with previous work demonstrating that injections of iE-DAP did not influence glucose tolerance in diet-induced obese mice [25]. L18-MDP’s stimulatory ability was further confirmed by our experiments on NCI-H716 cells which do not express *Gpr40* and *Gpr120* but exhibited a similar GLP-1 response to GLUTag cells [28,29,30]. The concentration of L18-MDP used in our cell studies is slightly below the amount of colonic luminal MDP recorded from a small healthy-human study (reported as 20–87 µM; Vavricka et al., 2004 [31]). It has also been suggested that under conditions of intestinal mucosal inflammation, MDP concentrations are likely to be much higher due to the rapid degradation of the peptidoglycan by phagocytic cells in the inflamed mucosa [32]. Nevertheless, the amount of luminal MDP that traverses the protective mucous layer and reaches the intestinal brush border as well as the mechanism by which MDP enters the L-cells has yet to be elucidated [33]. Future work with radiolabeled MDP may resolve this.

Next, we established the presence of MDP’s receptor, NOD2 in L-cells. Previous studies have localized NOD2 in the intestinal crypts and Paneth cells however, this is the first time NOD2 has been shown to be present in enteroendocrine cells [22,34]. While there were NOD2 positive cells outside of the L-cell population, nearly all of the GLP-1 positive cells were co-localized with NOD2. As the MAPK signaling pathway has been associated with both NOD2 activation [35] and GLP-1 secretion [36], we explored p38 MAPK phosphorylation during L18-MDP treatments. We observed a slight, but statistically significant increase in p38 MAPK phosphorylation after 10-min L18-MDP treatments but this increase was lost at the 20-min timepoint. However, we noted that the increase was modest and may not be of biological relevance. Although this provides some indication of mechanism, additional experiments are required to elucidate the full signaling cascade activated by L18-MDP. Future work should explore other NOD2 signaling cascades including extracellular regulated kinase (ERK) and c-Jun N-terminal kinase (JNK) [37].

In chow-fed animals, we found that MDP stimulated fasting GLP-1 levels only after two IP injections. This could be due to the clearance rate of this molecule as it takes about two hours for MDP to be almost completely excreted [25,38]. Therefore, the first injection may be priming the L-cells while the second injection results in significant GLP-1 release. In our experiments, even with the increase in GLP-1, we did not observe any improvements in insulin secretion or glucose tolerance. The inability of MDP to improve glucose clearance in chow-fed mice is in agreement with results previously described in MDP-treated euglycemic mice [25]. Interestingly, elevated GLP-1 levels were only observed in chow-fed mice and were lost after only 30 days on the HFD suggesting that phenotypic changes associated with the diet interfere with MDP’s action on L-cells.

While our goal was to evaluate the mechanism of MDP-improved glucose tolerance, we were unable to reproduce those previous findings [25]. Although we used the same strain of mice on the same 45% HFD and were injecting very similar or the same amounts of MDP, we could neither reach the same level of hyperglycemia even after a longer time period on the diet nor record any changes in glucose tolerance. This led us to repeat the experiment on mice fed a WD (which incorporates high sucrose in addition to high fat) to reach comparable fasting glucose levels. However, no improvements in MDP-induced glucose tolerance were observed. As this was a similar result to the HFD mice which had no alterations in GLP-1 or insulin, we did not further explore GLP-1 and insulin responses in the WD-fed mice. The discrepancy between our findings and previous studies could have occurred due to variations in blood collection techniques, time of fasting, time of year the study was conducted, animal handling, and facility factors. Nevertheless, we did observe a potential explanation for this loss in MDP sensitivity in the animals.

In L-cells, we determined that high glucose contributes to the down regulation of *NOD2* and *GCG* mRNA expression. A reduction in *NOD2* in L-cells during hyperglycemia may explain the loss of effect of MDP on GLP-1 in the obese mice. Consistent with our experiments, Grasset et al. (2017) [39] showed that the control of GLP-1 action on insulin secretion and gastric emptying was dramatically impaired in *NOD2* knock-out (KO) mice. Furthermore, the deletion of *NOD2* exacerbates diet-induced insulin resistance as NOD2 sensing of bacterial peptidoglycan provides protection from metabolic defects [40]. Thus, the peptidoglycan-NOD2 signaling pathway plays an important role in the control of GLP-1-induced insulin secretion and the inhibition of gastric emptying. Future work should analyze intestinal *NOD2* expression in HFD- and WD-fed mice to confirm our in vitro results.

In conclusion, MDP was only able to stimulate GLP-1 secretion in vitro when it was in its fatty acid modified form. The exact mechanism for L18-MDP stimulated GLP-1 release remains to be investigated but may act through the p38 MAPK signaling pathway. In vivo, we demonstrated that MDP stimulates fasting GLP-1 however, this effect is lost during the onset of hyperglycemia. Finally, we suggested that this loss of function is due to the down regulation of *NOD2* and *GCG* genes in hyperglycemic environments.

## 4. Materials and Methods

### 4.1. Animals

The animal use protocol (AUP) on this project is 3 September 2019. The approval date was 19 November 2019. Animal ethics were approved by the Laurentian University Animal Care Committee in accordance with the Canadian Council on Animal Care. Male and female C57BL/6 mice aged 6–7 weeks were purchased from Charles River Laboratories (St. Constant, QC, Canada). Mice were all co-housed in standard cages maintained on a 12 h light/dark cycle in the Paul Field Animal Care Facility at Laurentian University. All mice had food and water available *ad libitum* unless otherwise noted.

### 4.2. In Vivo MDP Experiments

#### 4.2.1. 45% High Fat Diet and Study Design

Mice were fed chow diet (8640, Envigo Teklad, Indianapolis, ID, USA) until 17 weeks in age when they were randomly divided into treatment and control groups. Mice received two intraperitoneal injections of MDP (5 mg/kg body weight; InvivoGen, San Diego, CA, USA) or phosphate-buffered saline (PBS) at time of fasting (16 h prior to oral glucose tolerance test (OGTT)) and 30 min prior to the OGTT. Dosing was chosen based on previous research [25,38]. After the initial experiment, mice were placed on a 45% High Fat Diet (D12451, Research Diets Inc., New Brunswick, NJ, USA). An OGTT was performed, and blood was collected and at 30, 60, and 100 days on the diet for glucose, insulin, and GLP-1 measurements.

#### 4.2.2. Western Diet and Study Design

Male and female mice, 6–7 weeks old were placed on the Western Diet (D12079B, Research Diets Inc., New Brunswick, NJ, USA) for 10 weeks. Mice received two intraperitoneal injections of MDP (5 mg/kg body weight; InvivoGen, San Diego, CA, USA) or PBS at time of fasting (16 h prior to OGTT) and 30 min prior to OGTT. For 3 day injections, mice received a once-daily intraperitoneal injection of MDP (100 µg/mouse) or PBS for three consecutive days and fasted 16 h prior to OGTT [25]. Blood glucose was measured at 0, 5, 30, 60, and 90 min as described.

### 4.3. Oral Glucose Tolerance Test (OGTT)

Mice were fasted overnight for 16 h in order to measure fasting glucose, GLP-1, and insulin. After fasting, animals received an oral glucose gavage (2 g/kg body weight in PBS). Ninety microliters of blood was collected into ethylenediaminetetraacetic acid (EDTA) coated tubes (Starstedt Inc, Germany) from the saphenous vein of the animal at 0, 15, and 60 min after the oral glucose challenge. An inhibitor cocktail (10% *v*/*v*) of Aprotinin (5000 KIU/mL; MilliporeSigma, Oakville, ON, Canada) and Diprotin A (0.1 mM; MilliporeSigma) was added to the blood sample prior to placing it on ice. Blood glucose was measured using a handheld glucometer (OneTouch Verio Flex) to assess glucose tolerance. Blood samples were stored on ice and centrifuged at 5000× *g*, 4 °C, for 5 min. Plasma was collected and stored at −20 °C (short-term) or −70 °C (long-term) for insulin and GLP−1 enzyme-linked immunosorbent assays (ELISA).

### 4.4. Evaluation of GLP-1 and Insulin Hormone Levels

Total GLP-1 and insulin levels in plasma samples were determined using mouse-specific ELISA kits (Crystal Chem, IL, USA) as per the manufacturer’s guidelines. For the GLP-1 ELISA, 10 µL of plasma diluted 1:5 in buffer solution was used for the assay. For the insulin ELISA, 5 µL of plasma was used per sample. Results are presented as absolute values for GLP-1 and change (delta) in insulin compared to 0-min timepoint for insulin.

### 4.5. In Vitro GLP-1 Secretion Experiments

Mouse GLP-1 secreting GLUTag cells (passage 10 to 30; contribution from Dr. Drucker, Lunenfeld-Tanenbaum Research Institute, Toronto, ON, Canada) and human GLP-1 secreting NCI-H716 (ATCC, Manassas, VA, USA) were used for these experiments. Unless otherwise stated, all cell culture media and reagents were obtained from MilliporeSigma. GLUTag cells were grown (37 °C, 5% CO_2_) in six-well plates (1,000,000 cells per well) or in 24-well plates (250,000 cells per well) in low glucose Dulbecco’s Modified Eagle Medium (DMEM) (10% fetal bovine serum (FBS), 1% penicillin/streptomycin (P/S)). NCI-H716 cells were grown (37 °C, 5% CO_2_) in 24-well plates (200,000 cells per well) coated with Matrigel (Corning Life Science, Tewksbury, MA, USA) in Roswell Park Memorial Institute medium (RPMI)-1640 (10% FBS, 1% P/S). Once the cells reached 80% confluency, growth media was replaced with secretion media (FBS reduced to 0.5%) containing vehicle or a range of doses (0.001–100 μg/mL) of NOD2 agonists, muramyl dipeptide (MDP) or 6-*O*-stearoyl-*N*-acetyl-muramyl-l-alanine-d-isoglutamine (L18-MDP) (InvivoGen) for a 2 h incubation period (37 °C, 5% CO_2_). Media was collected and acidified using trifluoroacetic acid (TFA) to a final concentration of 0.1% and stored at −20 °C until GLP-1 content was analyzed. GLP-1 was quantified using a multi-species enzyme-linked immunosorbent assay (ELISA) (BMS2194, Invitrogen, Thermo Fisher Scientific, Whitby, ON, Canada) as per the manufacturer’s guidelines. Briefly, GLUTag media samples were diluted 1:5 and NCI-H716 media samples were diluted 1:100 in assay buffer. Results for various treatments are presented as percent secretion relative to vehicle control. Cell viability was determined under similar experimental conditions using the neutral red uptake assay as described in Repetto’s work [41] where hydrogen peroxide (30%) was used as a positive control.

### 4.6. Gene Expression Analyses

GLUTag cells were grown in six-well plates at a density of 500,000 cells/mL for 24 or 48 h in low glucose (5 mM) or high glucose (25 mM) DMEM media (10% FBS, 1% P/S) at 37 °C, 5% CO_2_. After the incubation period, RNA was extracted from the cells with the BioRad (Ontario, Canada) total RNA kit following the manufacturer’s direction. The RNA was quantified with a UV spectrophotometer. cDNA was prepared using SensiFast cDNA synthesis kit (Bioline, ON, Canada) with 1 µg of RNA.

Primers were designed for the *NOD2* gene, *GCG* proglucagon gene, and the *RPL13a* reference gene and purchased from Integrated DNA Technologies (Coralville, IA, USA) (Table 1).

Quantitative real-time PCR reactions (20 µL) contained 0.8 µL of forward primer, 0.8 µL of reverse primer, 6.4 µL of RNase/DNase free water, 10 µL of 2× SensiFast SYBR No-ROX supermix (Bioline Inc., Taunton, MA, USA) and 2 µL of template cDNA that had been diluted in 20 µL of RNase/DNase free water. For NOD2 mRNA expression, the thermocycler had an initial denaturation step of 95 °C for 2 min, followed by 40 cycles of 95 °C for 5 s, 63 °C for 10 s and 72 °C for 20 s. For the GCG mRNA expression, the annealing temperature was 55 °C. PCR controls included a no-template control (NTC) reaction, replacing the cDNA with PCR grade water. Samples were run in triplicates and CT values were averaged. Results are shown as a ratio to RLP13a and normalized to control samples.

### 4.7. Fluorescent Immunohistochemistry

Formalin fixed paraffinized mouse ileum tissue sections of C57BL/6 mice were prepared on microscope slides. The tissue sections were deparaffinized as previously described in Gagnon’s work [42] excluding the microwave antigen retrieval. Briefly, tissues were blocked with TBS + 5% Normal Donkey Serum (NDS) (Abcam, Toronto, ON, Canada) for 20 min at room temperature (RT). Immediately after blocking, sections were incubated with primary antibodies 1:200 NOD2 (Novus Biologicals, CO, USA) and/or 1:250 GLP-1 (Abcam, Toronto, ON, Canada) and/or only blocking buffer overnight at 4 °C. After 3 × 5 min washes with TBS, the tissue samples were incubated for 45 min at RT with secondary antibodies 1:200 Donkey-anti-mouse 488 nm (Abcam, Toronto, ON, Canada) and 1:150 Goat-anti-rabbit 594 nm (Abcam, Toronto, ON, Canada) in blocking buffer. After 3 × 5 min washes with TBS, the samples were coated with Fluoroshield Mounting Medium with DAPI (Abcam, Toronto, ON, Canada) and a coverslip was fixed using clear nail polish. The tissues were observed in a dark room using an inverted Zeiss Axioplan fluorescent microscope and images were taken using Zeiss AxioVison software (Zeiss, Oberkochen, Germany). Semiquantitative analysis of NOD2 and GLP-1 expression was done using 4 images of different tissue sections at 20× magnification. All cells in the field of view were counted and results were presented as positive cells per 100 cells.

### 4.8. Western Blot

GLUTag cells were seeded into 6-well plates (1,000,000 cells per well) and grown for 48 h, as indicated above. On the day of the treatment, cells were rinsed with secretion media and then treated for 10 or 20 min with secretion media alone or secretion media containing 5 µg/mL L18-MDP at 37 °C, 5% CO_2_. Media was removed and wells were rinsed once with ice-cold PBS. Each well received 100 µL of lysis buffer (Cell Signaling Technologies, Danvers, MA, USA) containing protease and phosphatase inhibitors (Complete Mini/PhosphoSTOP; Roche LifeSciences, Guelph, ON, Canada). Cells were collected, sonicated on ice (10 s, Power 3), and centrifuged (5 min, 13,000× *g*, 4 °C). Protein concentrations were measured using the Bradford protein method and the expression levels of phospho-p38 MAPK and total p38 MAPK (1:1000, Cell Signaling Technologies, Danvers, MA, USA) were analyzed by Western blotting. BioRad Chemidoc XRS documentation apparatus was used to record the chemiluminescence signal and the signal intensity was measured using the QuantityOne program. Densitometry was used to analyze band intensity to compare phosphorylated p38 MAPK and total p38 MAPK protein expression between treatments.

To detect the NOD2 protein in NCI-H716 cells, cells were grown as previously described and treated after 48 h with RPMI 1640 secretion media for 2 h. Following the protocol described above, the NOD2 primary antibody (Novus Biologicals, Centennial, CO, USA) was diluted 1:1000 and the polyvinylidene fluoride (PVDF) membrane was treated as defined above.

### 4.9. Statistical Analyses

Values are reported as mean ± SEM. Experiments comparing two groups were analyzed using Student’s *t*-test. Experiments using several doses of a treatment were analyzed by one-way ANOVA followed by Dunnett’s post hoc test. Studies with two independent variables were analyzed by two-way ANOVA followed by a Sidak post hoc test. Statistical analysis was done using GraphPad Prism software. Values of n for each experiment are reported in the figure legends. *p* < 0.05 was considered significant.

## Figures and Tables

**Figure 1 ijms-21-05252-f001:**
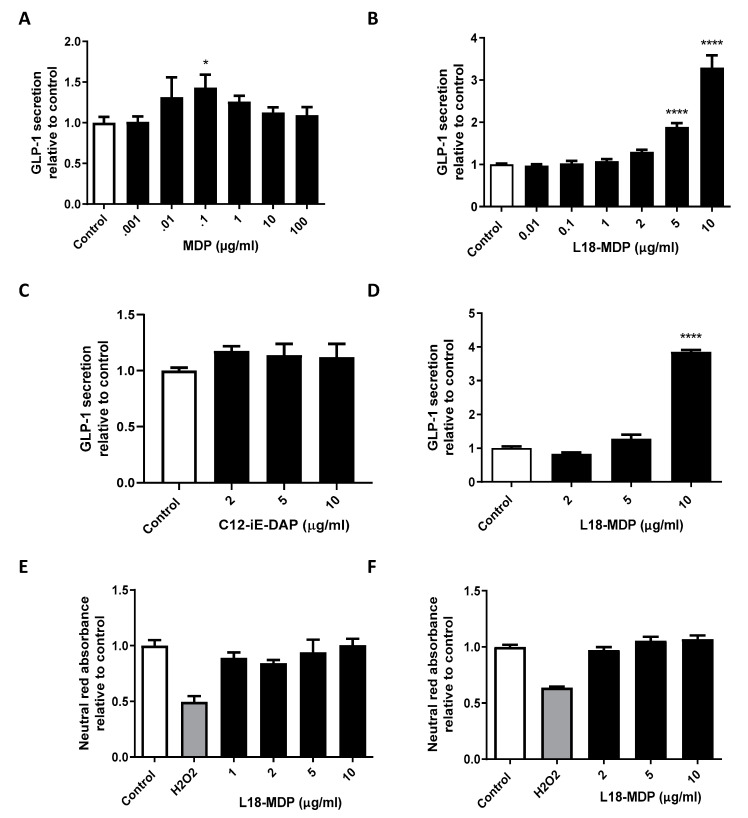
Fatty-acid modified muramyl dipeptide (L18-MDP) stimulates glucagon-like peptide-1 (GLP-1) secretion from L-cells. Percent active GLP-1 secretion relative to control was examined in GLUTag cells after 2 h treatments with MDP, *n* = 3–9 (**A**), L18-MDP, *n* = 3–18 (**B**), and fatty-acid modified γ-d-Glu-mDAP (C12-iE-DAP), *n* = 6 (**C**). Percent active GLP-1 secretion relative to control was examined in NCI-H716 cells after 2 h treatments with L18-MDP, *n* = 4 (**D**). Cell viability was measured by neutral red absorbance in GLUTag (**E**) and NCI-H716 (**F**) cells, *n* = 4. Results analyzed by one-way ANOVA and Dunnett’s post-hoc test. * *p* < 0.05, **** *p* < 0.0001 versus control cells. Values shown as mean ± SEM.

**Figure 2 ijms-21-05252-f002:**
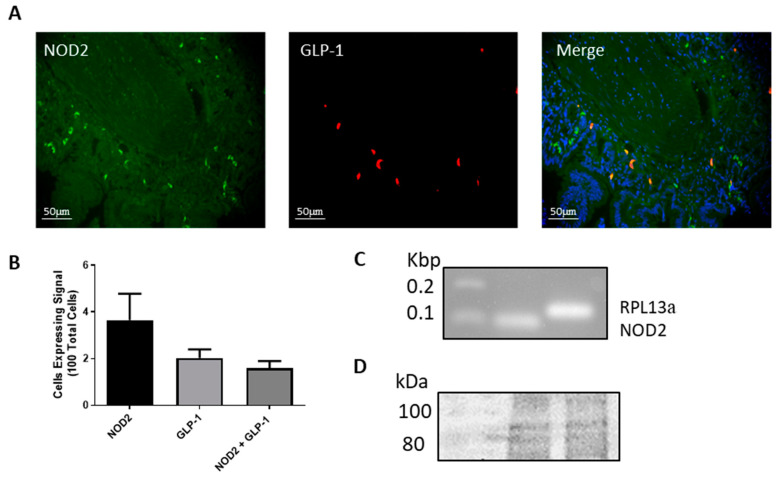
The nucleotide oligomerization domain 2 (NOD2) is expressed in L-cells. Ileum tissue was examined by fluorescent immunohistochemistry for NOD2 (green), GLP-1 (red) and nuclei (blue, in merge) (**A**). Cells expressing NOD2, GLP-1 or both were counted using 100 cells from four images to determine abundance within the ileum (**B**). mRNA expression of NOD2 and RLP13a was examined in GLUTag cells via RT-PCR (**C**). Protein expression of NOD2 in NCI-H716 cells via western blot (predicted size 97 kDa) (**D**).

**Figure 3 ijms-21-05252-f003:**
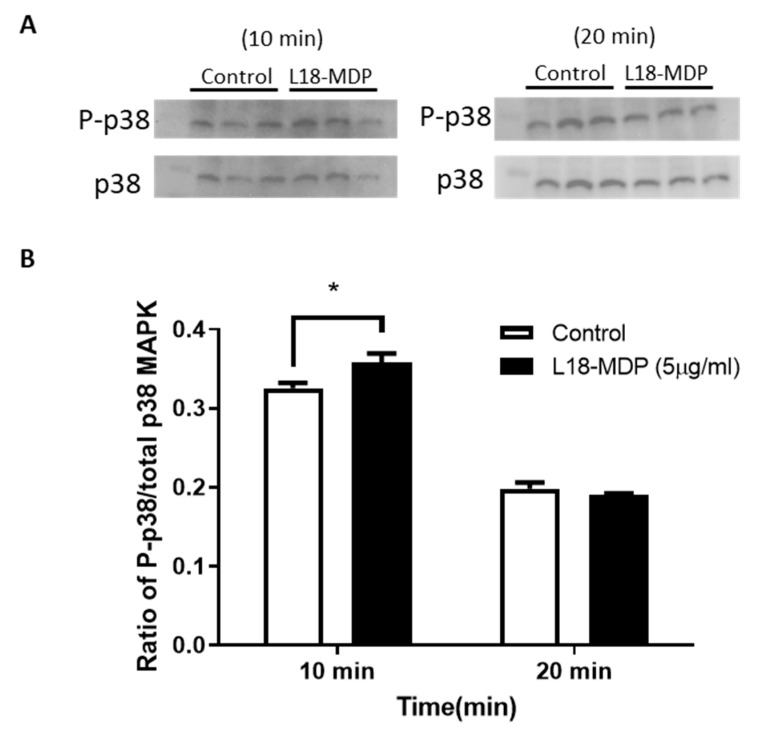
L18-MDP activates the mitogen-activated protein kinase (MAPK) signaling pathway in GLUTag cells after a 10 min exposure to L18-MDP. GLUTag cells were treated with 5 µg/mL L18-MDP for 10 or 20 min and phospho-p38 and total p38 MAPK were examined by western blot (**A**). Densitometry for P-p38/p38 was calculated (**B**), *n* = 3. Results were analyzed by two-way ANOVA and Sidak’s post-hoc test. * *p* < 0.05. Values shown as mean ± SEM.

**Figure 4 ijms-21-05252-f004:**
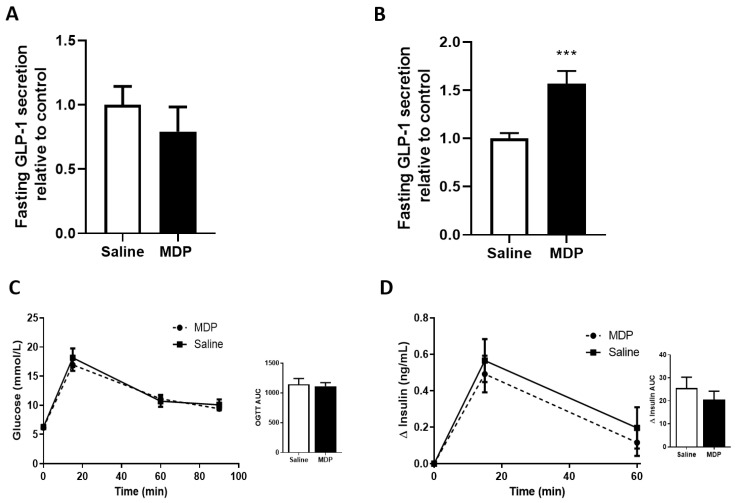
Two injections of MDP stimulates fasting GLP-1 in chow-fed mice. Fasting GLP-1 was measured after mice received one intraperitoneal (IP) injection of MDP (5 mg/kg) or saline, *n* = 2 mice per group (**A**). Fasting GLP-1 was measured after mice received two IP injections of MDP (5 mg/kg) or saline, *n* = 16–19 mice per group (**B**). Oral glucose tolerance test (OGTT) (2 g/kg) of mice treated with two IP injections of MDP (5 mg/kg) or saline (**C**) and insulin analysis (**D**). *n* = 7–10 mice per group. Results were analyzed by one-way ANOVA or unpaired *t*-test where appropriate. *** *p* < 0.001. Values shown as mean ± SEM.

**Figure 5 ijms-21-05252-f005:**
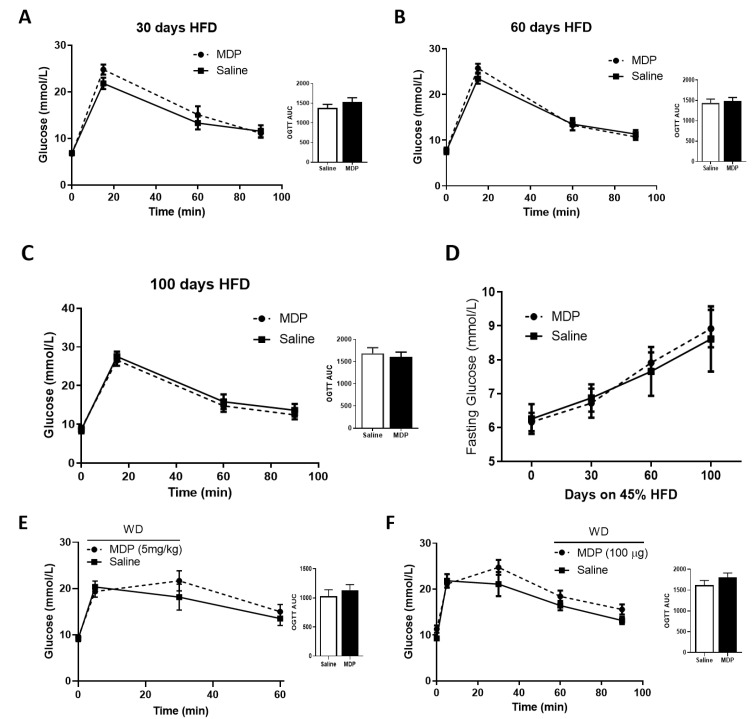
MDP has no effect on glucose clearance in mice fed obesogenic diets. Oral glucose tolerance test (2 g/kg) (**A**–**C**) and fasting blood glucose (**D**) were measured after 30 days (**A**), 60 days (**B**) and 100 days (**C**) on a 45% high-fat diet (HFD) after mice received two IP injections of MDP (5 mg/kg) or saline. *n* = 7–10 mice per group. Oral glucose tolerance test (2 g/kg) after 70 days on Western Diet with two IP injections of MDP (5 mg/kg) or saline, *n* = 8–12 mice per group (**E**). OGTT in male mice after 110 days on Western Diet with three IP injections of MDP (100 μg) or saline, *n* = 4–6 mice per group (**F**). Results were analyzed by one-way ANOVA or unpaired *t*-test where appropriate. Values shown as mean ± SEM.

**Figure 6 ijms-21-05252-f006:**
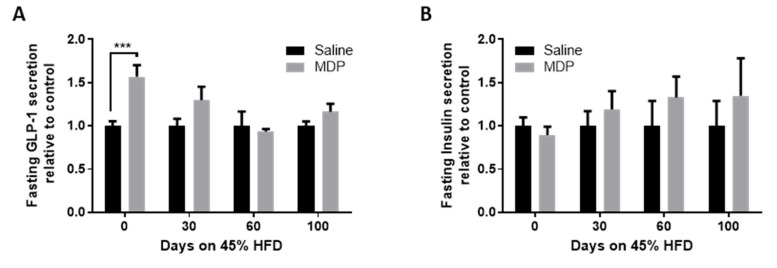
Elevated fasting GLP-1 is lost with the development of hyperglycemia. Fasting GLP-1 (**A**) and insulin (**B**) were measured after mice received two IP injections of MDP (5 mg/kg) or saline, *n* = 7–19 mice per group. Results analyzed by two-way ANOVA with Sidak post-hoc test. *** *p* < 0.001. Values shown as mean ± SEM.

**Figure 7 ijms-21-05252-f007:**
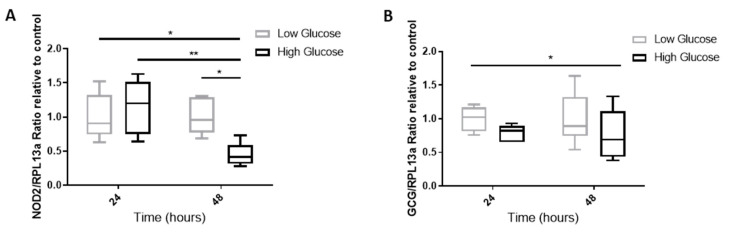
*NOD2* and *GCG* expression are reduced under high glucose conditions. GLUTag cells were incubated for 24 or 48 h in low (5 mM) or high (25 mM) glucose media. Expression for *NOD2* (**A**) and *GCG* (**B**) were examined by RT-qPCR and analyzed using a two-way ANOVA and Sidak post-hoc test. * *p* < 0.05, ** *p* < 0.01, *n* = 6.

**Table 1 ijms-21-05252-t001:** Primer sequences used for gene analysis in murine GLUTag cells.

Gene	Forward 5′-3′	Reverse 5′-3′
*NOD2*	5′-GTCCAACAATGGCATCACCT-3′	5′-TGTGTTCCCTCGAAGCCAAA-3′
*GCG*	5′-TTGAGAGGCATGCTGAAGGG-3′	5′-TCTTCTGGGAAGTCTCGCCT-3′
*RPL13a*	5′-GAAGCAGATCTTGAGGTTACGGA-3′	5′-AGGCATGAGGCAAACAGTCT-3′

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
