# Peer review of "The Role of the Bacterial Muramyl Dipeptide in the Regulation of GLP-1 and Glycemia"

_ijms, 2020, doi:10.3390/ijms21155252_

Round 1

Reviewer 1 Report

The authors have investigated the role of MDP in regulating GLP-1 levels. They show that an acylated MDP, L18-MDP, but not MDP, induces secretion of GLP-1in GLUTag and NCI-H716 cells. They further show that 2 injections of MDP induces increased secretion of GLP-1 in mice maintained on chow but not in mice maintained on high fat diet or western diet. They show that GLP-1 expressing cells also express NOD2 and expression of Nod2 was decreased in cells maintained in high glucose medium. The remaining data either shows no difference or the difference in not biologically significant.

The authors have interesting results; however, they fail to connect the data in the different figures and do not present a cohesive story. This reviewer has the following suggestions, which would improve the manuscript:

Fig. 1 – L18-MDP, but not MDP, induces GLP-1 secretion. The significance of this data is questionable as L18-MDP does not occur naturally and therefore cells in the host would not be naturally exposed to it either from the gut bacteria or from a pathogen. L18 should be used as a control for this experiment. Is the increase in GLP1 accompanied with an increase in insulin in these cell lines?

Fig. 3 – This data should not be included in the manuscript as the observed changed in phosph-p38 is likely not biologically significant. Why did the authors not look at other MAPKs, ERKs and JNK?

Fig. 6 – The authors investigate the in vivo effect of HFD on GLP-1. They should also determine the effect of the Western diet on GLP-1 secretion.

Fig. 7 – What is the effect of the HFD and WD on expression of Nod2.

Author Response

The authors have investigated the role of MDP in regulating GLP-1 levels. They show that an acylated MDP, L18-MDP, but not MDP, induces secretion of GLP-1in GLUTag and NCI-H716 cells. They further show that 2 injections of MDP induces increased secretion of GLP-1 in mice maintained on chow but not in mice maintained on high fat diet or western diet. They show that GLP-1 expressing cells also express NOD2 and expression of Nod2 was decreased in cells maintained in high glucose medium. The remaining data either shows no difference or the difference in not biologically significant.

The authors have interesting results; however, they fail to connect the data in the different figures and do not present a cohesive story. This reviewer has the following suggestions, which would improve the manuscript:

-We appreciate this feedback and have added a summary paragraph to the discussion to help the cohesiveness (page 8, lines 21-28).

Fig. 1 – L18-MDP, but not MDP, induces GLP-1 secretion. The significance of this data is questionable as L18-MDP does not occur naturally and therefore cells in the host would not be naturally exposed to it either from the gut bacteria or from a pathogen.

-We agree, the acylated form of MDP used is not the naturally occurring form of this muropeptide. We selected this modified form as GLP-1 secretion from cell lines is assessed over a 2 hour period, and the uptake of the unmodified form may be insufficient during that time period. We have provided some references to clarify the use of the acylated form and we have added some additional discussion to acknowledge this limitation (page 9, lines 3-6).

 L18 should be used as a control for this experiment.

-This would be an ideal control, however was not available from the muropeptide vendor. To partially control for the prevalence of a fatty acid we incorporated a separate acylated treatment (C12-iE-DAP) which had no effect on GLP-1 secretion. We have provided some explanation of this in the discussion (page 9, lines 7-10).

Is the increase in GLP1 accompanied with an increase in insulin in these cell lines?

-As the cells used (GLUTag and NCI) are monoclonal GLP-1 secreting cells, their insulin production would be minimal if all. A potential new experiment would be to treat beta cell lines with MDP to observe an effect on insulin.  

Fig. 3 – This data should not be included in the manuscript as the observed changed in phosph-p38 is likely not biologically significant. Why did the authors not look at other MAPKs, ERKs and JNK?

-We agree the statistical significance in the p38 MAPK Western was borderline and likely not biologically significant. Nevertheless, we felt it was important to demonstrate that a kinase normally associated with NOD2 signaling and GLP-1 release was not strongly affected. We have clarified this in the discussion and indicated that future work will explore the cellular pathways in detail including those listed above (page 9, lines 27-33).

Fig. 6 – The authors investigate the in vivo effect of HFD on GLP-1. They should also determine the effect of the Western diet on GLP-1 secretion.

-We agree that adding a GLP-1 analysis in the Western diet could add value to the study, However, as we were not able to achieve any impact on blood glucose we deemed any change in GLP-1 to be unlikely. We have noted this limitation in the discussion (page 9, lines 47-49).

Fig. 7 – What is the effect of the HFD and WD on expression of Nod2.

-This is a very important experiment and intestinal tissue was collected at the end of the study. Unfortunately due to Covid-19 restrictions, our lab access was restricted. Nevertheless, we have added the acknowledgment and this limitation to the discussion (page 10, lines 8-9).

Reviewer 2 Report

In this original science basic article, Authors showed that muramyl dipeptide (MDP), a component of bacterial cell wall, may increase the production of GLP1 in human L cells, with a pathway dependant on NOD2. In mice, MDP was able to increase GLP1 levels, although without improving glucose tolerance. Finally, a high-fat diet counteracted the beneficial effects of MDP.

This is a good paper with nicely planned experiments, simple and easy to read. I have only few minor observations:

1) In the Discussion, Authors should underline that a high fat diet of only 30 days was able to impair GLP1 secretion, as shown in figure 6.

2) NOD2 is not only involved in regulation of glucose tolerance. It has been elucidated that it is a pivotal driver of microbiota composition (see Daniel N et al, Cell Mol Gastroenterol Hepatol 2020). Please discuss.

Author Response

In this original science basic article, Authors showed that muramyl dipeptide (MDP), a component of bacterial cell wall, may increase the production of GLP1 in human L cells, with a pathway dependant on NOD2. In mice, MDP was able to increase GLP1 levels, although without improving glucose tolerance. Finally, a high-fat diet counteracted the beneficial effects of MDP.

This is a good paper with nicely planned experiments, simple and easy to read. I have only few minor observations:

1) In the Discussion, Authors should underline that a high fat diet of only 30 days was able to impair GLP1 secretion, as shown in figure 6.

-point added (page 9, lines 40-42)

2) NOD2 is not only involved in regulation of glucose tolerance. It has been elucidated that it is a pivotal driver of microbiota composition (see Daniel N et al, Cell Mol Gastroenterol Hepatol 2020). Please discuss.

-We agree and have added this point and reference to the introduction (page 2, lines 18-19).

Round 2

Reviewer 1 Report

This reviewer had suggested some additional experiments to make the manuscript stronger. However, the authors are unable to perform these experiments due to lack of access to their lab. We can accept the manuscript in its present form.